

An Improved BRDF Hotspot Model and its Use in VLIDORT to Study the Impact
of Atmospheric Scattering on Hotspot Directional Signatures in the Atmosphere
Xiaozhen Xiong[1*], Xu Liu[1], Robert Spurr[2],
Ming Zhao[1,3], Qiguang Yang[1,3], Wan Wu[1], Liqiao Lei[1,3]
[1] NASA Langley Research Center, Hampton, VA, USA
[2] RT SOLUTIONS Inc., Cambridge, MA, USA
[3] Adnet Systems Inc., Bethesda, MD 20817, USA
Corresponding to: Xiaozhen Xiong (Xiaozhen.Xiong@nasa.gov)
**Abstract**
The term "hotspot" refers to the sharp increase of reflectance occurring when incident (solar) and
reflected (viewing) directions coincide in the backscatter direction. The accurate simulation of
hotspot directional signatures is important for many remote sensing applications. The RossThick-
LiSparse-Reciprocal (RTLSR) Bidirectional Reflectance Distribution Function (BRDF) model is
widely used in radiative transfer simulations, but it typically requires large values of numerical
quadrature and Fourier expansion terms in order to represent the hotspot accurately. In this paper,
we have developed an improved hotspot BRDF model that converges much faster, making it more
practical for use in atmospheric radiative transfer simulations of top-of-atmosphere (TOA) hotspot
signatures. Using the VLIDORT RT model, we found that reasonable TOA hotspot accuracy can
be obtained with just 23 Fourier terms for clear atmospheres, and 63 Fourier terms for atmospheres
with aerosol scattering.
We carried out a number of hotspot signature simulations with VLIDORT to study to the impact
of molecular and aerosol scattering on hotspot signatures. We confirmed that (1) atmospheric
scattering tends to smooth out the hotspot signature at the TOA, but has no impact on hotspot
width; and (2) the hotspot signature at the TOA in the near-infrared is larger than in the visible,
and has an obvious increase with the solar zenith angle. As the hotspot amplitude at the TOA with
aerosol scattering included is smaller than that with molecular scattering only, the amplitude of
hotspot signature at the surface is likely underestimated in the previous analysis based on the
POLDER measurements, where the atmospheric correction was based on a single-scatter
Rayleigh-only calculation. We also draw attenuation to a scaling factor of $3\pi/4$ which has been
applied to the Ross-Thick kernel with hotspot correction.
Keywords: BRDF, Hot Spot, VLIDORT, RTLSR






## 1. Introduction

Most land surfaces reflect incident light anisotropically. For a given incident sun angle, the surface reflectance may vary by a factor of two in the near infrared [Kriebel et al., 1978]. An accurate accounting of the anisotropic reflectance at the Earth's surface is very important for many remote sensing applications, including monitoring of climate changes, mapping land covers, analyzing vegetation densities, or inter-calibration between different satellite instruments (e.g. [Yang et al., 2020] and references therein). Lorente et al. [2018] investigated the importance of surface reflectance anisotropy with regard to cloud and $NO_2$ retrievals from satellite measurements by the Global Ozone Monitoring Experiment 2 (GOME-2) and the Ozone Monitoring Instrument (OMI). This study showed that retrieved cloud fractions have an east–west across-track bias of 10-50 %, and under moderately polluted $NO_2$ scenarios with backward scattering geometry, clear-sky air mass factors can be as much as 20% higher when surface anisotropic reflection is included in the calculations.

The angular distribution of reflected light by a surface is normally represented mathematically by the Bidirectional Reflectance Distribution Function (BRDF) [Nicodemus et al., 1992], which is a function of the incident solar zenith angle, the reflected viewing zenith angle, and the relative azimuth angle between these two directions. Usually, there is a strong increase in BRDF toward the backscatter direction, with much smaller BRDF variations seen around the forward-scatter direction. Peak BRDF values occur when backscatter incident and reflected directions coincide; this sharp reflectance increase is usually referred to as the "hotspot" [Kuusk, 1985; Hapke, 1986]. The "hotspot" effect has been observed for a variety of planetary bodies, including the Moon, Mars, asteroids, planetary satellites, as well as terrestrial vegetation [Bréon et al., 2002]. The most widely accepted explanation for the hotspot effect is the so-called "shadow hiding" effect. Here, particles at the surface (e.g. leaves, soil grains) cast shadows on adjacent particles; these shadows are visible at large phase angles but at zero phase angle the shadows are hidden by the particles that cast them. Coherent backscatter is another physical explanation of reflectance enhancement in the hotspot direction [Kuga and Ishimaru, 1984; Hapke et al., 1993].

The bidirectional reflective spectra of land surfaces have been measured in laboratories, fields and airborne experiments, or derived from satellite observations. The two most widely used hyperspectral bidirectional reflective spectra of land surfaces are (1) the U.S. Geological Survey (USGS) Spectral Library (Version 7) [Kokaly et al., 2017], comprising a very diverse land surface BRDF data based with about 40,000 spectra in all, and (2) the ASTER Spectral Library from NASA's Jet Propulsion Laboratory, with a collection of over 2,000 measured spectra [Baldridge et al., 2009]. Using these two databases and RTLSR model, Yang et al. [2020] went on to develop a Hyper-Spectral Bidirectional Reflectance (HSBR) model for remote sensing applications. BRDF data derived from satellite observations have been used to evaluate and correct for anisotropy in several instruments, including, for example, the Advanced Very High Resolution Radiometer (AVHRR) [e.g. Gutman, 1987; Roujean et al., 1992], the Along-Track Scanning Radiometer (ATSR-2) located on board on the ERS-2 platform [Godsalve, 1995], and the MODerate resolution Imaging Spectrometer (MODIS) [Wanner et al., 1997; Lucht et al., 2000; Schaaf et al., 2002]. However, the AVHRR, ATSR and MODIS instruments have limited viewing geometry options; in contrast, the POLarization and Directionality of Earth Reflectances (POLDER) instrument on board the Advanced Earth Observing Satellite (ADEOS) in August 1996 provided a much better



directional sampling to measure the BRDF up to 65° VZA (viewing zenith angle) and for the full
azimuth range [Deschamps et al., 1994]. So, these POLDER reflectance measurements were used
to examine the hotspot signature for different vegetated surfaces [Bréon et al., 2002].
Many BRDF models have been developed in order to simulate or reproduce directional signatures
of land surface reflectance. These include empirical models [Walthall et al., 1985], semi-empirical
models [Hapke, 1981, 1986; Rahman et al., 1993; Roujean et al., 1992; Wanner et al., 1995; 1997;
Lucht et al., 2000], and physical models [Pinty and Verstraete, 1991]. In particular, kernel-driven
semi-empirical models have been used frequently to generate global BRDF and albedo products.
Several studies have identified the so-called Ross-Thick-Li-Sparse-Reciprocal (hereinafter
"RTLSR") kernel combination as the BRDF model best suited for the operational MODIS
BRDF/Albedo algorithm [Wanner et al., 1997; Lucht et al., 2000; Schaaf et al., 2002]. Using about
22,000 sets of the measured BRDFs derived from carefully selected cloud-free measurements with
large directional coverage from the spaceborne POLDER instrument [Bicheron and Leroy, 2000],
Maignan et al. [2004] evaluated the efficacy of several analytical models to reproduce these
observed BRDF signatures. They found that a simple kernel-driven model with only three free
parameters can provide an accurate representation of the BRDF. One of the best such models is
the three-parameter linear Ross–Li model. However, this model fails to capture the sharp
reflectance increase centered around the hotspot backscatter direction. From an analysis of
POLDER data, a correction to this model to capture the hotspot effect was proposed by [Bréon et
al., 2002]. By means of an explicit representation of the hotspot effect for a few degrees around
the backscattering direction, Maignan et al. [2004] found that the hot-spot modified RTLSR linear
BRDF model with three free parameters produced the best agreement with measurement. This
BRDF model from [Maignan et al., 2004] was referred to as the "Ross–Li–Maignan" model in
[Vermont et al., 2009].
With three linear parameters characterizing the Ross–Li model, it is a straightforward process to
invert the model by minimizing the Root Mean Square (hereafter RMS) difference between the
measurements and the modeled directional reflectances. This BRDF inversion technique has been
used to derive the MODIS BRDF/Albedo product [Schaaf et al., 2002]. An improvement was made
by Vermont et al. (2009) to correct the time series of surface reflectance derived from MODIS.
Using POLDER data, Bacour and Bréon [2005] retrieved the three parameters, using the modified
Ross-Li model, and further analyzed the variability of these parameters with vegetation cover
types. A common approach to derive the surface reflectance directional signatures from satellite
observations is to first remove the atmospheric absorption and scattering effects. This process,
which converts the top of the atmosphere (TOA) signal to a surface reflectance, is often called
"atmospheric correction". The surface is generally taken to be Lambertian in such atmospheric
correction algorithms; however, it was found that without considering the BRDF effects,
atmosphere correction errors can reach up to 10% at certain geometries and under turbid conditions
[Vermote et al., 1995]. Since the mid-1980s, atmospheric correction algorithms have evolved from
the earlier "empirical line" and "flat-field" methods to more modern approaches based on rigorous
radiative transfer modeling [Gao et al., 2009]. Clearly, the accurate simulation of atmospheric and
surface radiative transfer is a critical element in the derivation of surface BRDF from satellite
measurements.
Several key numerical radiative transfer models (RTMs) were developed in the 1980s, and the
most popular RTMs in use today are usually based on discrete ordinate methods or the doubling-
adding technique. Following detailed mathematical studies made by Hovenier and others





[Hovenier and van der Mee, 1983; de Rooij and van der Stap, 1984], a general doubling-adding
model was developed for atmospheric radiative transfer modeling, e.g. [de Haan et al., 1987;
Stammes et al., 1989]. DISORT is a discrete ordinate model developed by Stamnes and co-
workers and released for public use in 1988 [Stamnes et al., 1988; Stamnes et al., 2000]; a vector
discrete ordinate model (VDISORT) was developed later on in the 1990s [Schulz et al., 1999]. In
the 1980s, Siewert and colleagues made a number of detailed mathematical examinations of the
vector RT equations. The development of the scattering matrix in terms of generalized spherical
functions was reformulated in a convenient analytic manner [Siewert, 1981; Siewert, 1982;
Vestrucci and Siewert, 1984], and a new and elegant solution from a discrete ordinate viewpoint
was developed for the scalar [Siewert, 2000a] and vector [Siewert, 2000b] single-layer slab
models. LIDORT [Spurr et al., 2001; Spurr, 2002] and VLIDORT [Spurr, 2006] are multiple-
scattering multi-layer discrete ordinate scattering codes with simultaneous linearization facilities
for the generation of the radiation field and analytically-derived Jacobians (weighting functions or
partial derivatives of the radiation field with respect to any atmospheric or surface parameter).
SCIATRAN is a comprehensive software package for the modeling of radiative transfer processes
in the terrestrial atmosphere and ocean from the ultraviolet to the thermal infrared, including
multiple scattering processes, polarization, thermal emission and ocean–atmosphere coupling; the
software package contains several radiative transfer solvers including discrete-ordinate techniques
[Rozanov et al., 2014]. The Second Simulation of the Satellite Signal in the Solar Spectrum (6S)
[Vermote et al., 1997] RTM is widely used in the atmospheric correction community; 6S is based
on the successive orders of scattering approach (SOS) [Lenoble et al., 2007]. In this study, we will
use the VLIDORT RTM, which has a fully-developed supplemental code package for the
generation of surface BRDFs. This supplement includes a variety of BRDF kernel models (semi-
empirical BRDF functions developed for particular types of surfaces) that can be combined
linearly to that provide total BRDFs required as input for the full VLIDORT RTM calculations.
These kernels include the Ross–Li model both with and without the hotspot correction.
In the first part of this study (Section 2) we discuss the Ross-Li kernel hotspot correction in detail,
and present an alternative model of the hotspot correction; this new formulation is designed to
improve the hotspot convergence with respect to the number of cosine-azimuth Fourier terms
needed to represent the BRDF and also to the number of azimuth quadrature angles needed for the
numerical derivation of these Fourier terms. In Section 3, we investigate accuracies for
reconstructed BRDFs in the hotspot region, comparing our new model with older hot-spot
corrections. Then, using VLIDORT and the new hotspot correction model, we examine the impact
of atmospheric scattering on the simulated TOA-hotspot signature. Summary and conclusions are
given in Section 4.



## 2. Hotspot BRDF Models

### 2.1.    RossThick-LiSparse-Reciprocal (RTLSR) BRDF model

Land surfaces possess complicated structural elements, making the reflective properties of such surfaces very hard to model. The geometric structure of a given land surface greatly influences its reflectance, thanks to shadowing and multiple scattering effects [Roujean et al., 1992]; this angle-dependent scattering component is called "geometric scattering". Another structure-related scattering effect is called "volumetric scattering", which usually consists of multiple reflections from different components within a volume and produces a minimum reflectance near nadir viewing. Scattering by trees, branches, soil layers, and snow layers are typical manifestations of volumetric scattering. These two scattering processes are usually used to characterize the surface BRDF. For example, the operational Moderate Resolution Imaging Spectroradiometer (MODIS) BRDF/Albedo product is derived based on semi-empirical kernel-driven linear BRDF models that composes of three components: an isotropic scattering term, a geometric scattering kernel, and a volumetric scattering kernel. The RossThick-LiSparse-Reciprocal (RTLSR) kernel combination has been identified as the best model suited for the operational MODIS BRDF/Albedo retrieval ([Schaaf et al., 2002] and references therein), in which the land surface reflectance function $B(\theta_i, \theta_r, \Delta\varphi)$ is represented as:

$$B(\theta_i, \theta_r, \Delta\varphi) = P_1 K_{Lamb} + P_2 K_{geo}(\theta_i, \theta_r, \Delta\varphi, P_4, P_5) + P_3 K_{vol}(\theta_i, \theta_r, \Delta\varphi). \quad (1)$$

Here, $\theta_i$ and $\theta_r$ are the incident (solar) and reflected (viewing) zenith angles, and $\varphi_i$ and $\varphi_r$ the corresponding azimuth angles, with $\Delta\varphi = \varphi_r - \varphi_i$ the relative azimuth angle. $P_1$ is the Lambertian kernel amplitude with $K_{Lamb} \equiv 1$, while $P_2$ and $P_3$ are the weights of the Li-Sparse-Reciprocal geometric scattering kernel $K_{geo}$ and the Ross-Thick volume scattering kernel $K_{vol}$ respectively. Parameters $P_4$ and $P_5$ characterize $K_{geo}$ and are discussed below. This 3-kernel semi-empirical model has shown surprising ability to reproduce with high accuracy the measured directional signatures of the main land surfaces; the RTLSR model is significantly better than other analytical models or combinations thereof [Maignan et al., 2004]

The Li-Sparse-Reciprocal geometric scattering kernel was derived from surface scattering and the theory of geometric shadow casting by [Li and Strahler, 1992], and is given by

$$K_{geo}(\theta_i, \theta_r, \Delta\varphi, P_4, P_5) = \frac{1 + sec\theta_r' sec\theta_i' + tan\theta_r' tan\theta_i' cos\Delta\varphi}{2} +$$

$$\left(\frac{t - sint cost}{\pi} - 1\right)(sec\theta_r' + sec\theta_i'). \quad (2)$$

$$cos^2 t = \left(\frac{P_4}{sec\theta_r' + sec\theta_i'}\right)^2 [G(\theta_r', \theta_i', \Delta\varphi)^2 + (tan\theta_r' tan\theta_i' sin\Delta\varphi)^2]; \quad (3)$$

$$G(\theta_r', \theta_i', \Delta\varphi) = \sqrt{tan^2\theta_r' + tan^2\theta_i' - 2tan\theta_r' tan\theta_i' cos\Delta\varphi}; \quad (4)$$

$$tan\theta_r' = P_5 tan\theta_r; \qquad tan\theta_i' = P_5 tan\theta_i. \quad (5)$$

We note also the following expression for the scattering angle $\zeta$:

$$cos\zeta = cos\theta_r cos\theta_i + sin\theta_r sin\theta_i cos\Delta\varphi \quad (6)$$

Assuming a dense leaf canopy, and tree crowns that are spheroids with vertical length $2b$, horizontal width $2r$, and centroid distance $h$ above the ground, then $P_4 = h/b$ and $P_5 = b/r$ are



two parameters representing the crown relative height. $P_4$ and $P_5$ can be obtained empirically,
and they are usually assumed to take values 2 and 1 respectively.
The Ross-Thick volume scattering kernel $K_{vol}$ was derived from volume scattering radiative
transfer models by [Ross, 1981], and it is often referred to as "*Ross thick*" [Wanner et al., 1995]:
$$K_{vol}(\theta_i, \theta_r, \Delta\varphi) = \frac{(\frac{\pi}{2} - \zeta)\cos\zeta + \sin\zeta}{\cos\theta'_r + \cos\theta'_i} - \frac{\pi}{4}. \tag{7}$$

Since we are using the RTLSR linear model to reproduce natural target BRDFs, it follows that the
three parameters will contain most of the reflectance directional information for view angles of
less than 60°. Theoretically, parameter $P_1$ and $P_2$ in Eq. (1) can be derived, but due to the extensive
variability of surface cover and biome types, there remains the practical question as to the
determination of the free parameters [Vermont et al., 2009], and for the MODIS BRDF/Albedo
product, $P_1$, $P_2$ and $P_3$ are derived from MODIS measurements in a few channels. A hyperspectral
bidirectional reflectance (HSBR) model for land surface was developed by [Yang et al., 2020].
The HSBR model includes a diverse land surface BRDF database with about 40,000 spectra, stored
in terms of the three Ross-Li parameters. The HSBR model has been validated using the USGS
vegetation database and the AVIRIS reflectance product, and can be used to generate hyperspectral
reflectance spectra at different sensor and solar observation geometries.

### 2.2. *Hot-Spot models, including an improved formulation*

Based on an analysis of POLDER measurements, Bréon et al.[2002] found that the hotspot
directional signature is proportional to $\left(1 + \zeta/\zeta_0\right)^{-1}$, where $\zeta_0$ is the hotspot halfwidth that can
be related to the ratio of scattering element size and canopy vertical density. This hotspot modeling
has been validated against measurements acquired with the spaceborne POLDER instrument with
a very high directional resolution, i.e. on the order of 0.3° [Bréon et al., 2002]. Maignan et al.[
2004] brought this hotspot correction into the Ross-Li model, and re-wrote the Ross thick kernel
with hotspot correction as:
$$K_{vol} = \frac{4}{3\pi} \frac{(\frac{\pi}{2} - \zeta)\cos\zeta + \sin\zeta}{\cos\theta'_r + \cos\theta'_i}(1. + \frac{1}{1 + \zeta/\zeta_0}) - \frac{1}{3}. \tag{8}$$

We note here that there is a difference of a factor of $\frac{4}{3\pi}$ between Eqs. (7) and (8). Bréon et al. [2002]
indicated that $\zeta_0$ is generally in a small range between 0.8° to 2°, while some dispersion occurs in
the range 1°–4° for scenarios classified as forest and desert types in the International Geosphere-
Biosphere Program (IGBP) system. For the sake of simplicity, and to avoid the addition of a free
parameter in the BRDF modeling, Maignan et al.[2004] suggested setting a constant value of $\zeta_0 =$
1.5°. The version of the RTLSR model which accounts for the hotspot signature using Eq. (8) will
be denoted as RossThickHT-M in this paper. Using multidirectional PARASOL (Polarization &
Anisotropy of Reflectances for Atmospheric Sciences coupled with Observations from a
Lidar) data at coarse resolution (6 km) over a large set of representative targets, Maignan et
al.[2004] showed that the simple three-parameter model permits accurate representation of the
BRDFs.
Another hotspot correction was developed by Chen and Cihlar[1997] as a negative exponential
function, and Jiao et al. [2013] brought this latter correction to the Ross-Li model, as follows:





$$K_{vol} = \frac{4}{3\pi} \frac{\left(\frac{\pi}{2}-z\right)\cos\zeta + \sin\zeta}{\cos\theta'_r + \cos\theta'_i} (1 + C_1 e^{\left(-\frac{\zeta}{\pi}\right)C_2}) - \frac{1}{3}.$$ 
(9)

Here, $C_1$ is physically related to the difference between the spectral reflectance of foliage and the
background, controlling the height of the hotspot; $C_2$ is related to the ratio of canopy height to the
size of the predominant canopy structure, determining the width of the hotspot. We found that we
can simply set $C_2$ to be $\zeta_0$. We remark that $\zeta_0$ is given in radians in Eq. (8) and in degrees in Eq.
(9). However, Bréon et al.[2002] determined that observed hotspot signatures are better fitted with
a function of $\left(1 + \zeta/\zeta_0\right)^{-1}$ rather than with a negative exponential that is often used for hotspot
modeling.
In this paper, we denote the version of the RTLSR model that accounts for the Hot-Spot process
using Eq. (9) as RossThickHT-C. Some validation to the RossThickHT-C model has been made
by Jiao et al. [2013]. Although one advantage of RossThickHT-C model is the ability to use
parameter $C_1$ to adjust the amplitude of hotspot [Jiao et al., 2013], such an adjustment can be also
easily made by adding one parameter in the correction term in Eq. (8), i.e. to change $\left(1 + \zeta/\zeta_0\right)^{-1}$
to $C_1/(1 + \zeta/\zeta_0)$. With this in mind, our effort will focus on an improvement in the Ross-Thick
BRDF kernel, starting with the baseline model of Maignan et al. [2004].
A number of kernel BRDF models have been incorporated in the LIDORT and VLIDORT RTMs,
including the RTLSR model and the RossThickHT-M model. In VLIDORT (and this applies
equally to other polarized radiative transfer models). it is necessary to develop solutions for the
radiation fields in terms of Fourier cosine and sine azimuth series; the same considerations apply
to the BRDFs. For scalar kernel models without polarization, only the Fourier cosine series is
needed. The Fourier components of the total BRDF are calculated through:
$$B^m(\mu, \mu') = \frac{1}{2\pi} \int_0^{2\pi} B(\mu, \mu', \varphi) \cos m\varphi \, d\varphi \,.$$ 
(10)

Integration over the azimuth angle is done by double numerical quadrature over the ranges [0, $\pi$]
and [$-\pi$, 0]. The number of BRDF azimuth quadrature abscissa ($N_{BRDF}$) should be set to at least
100 in order to obtain a numerical accuracy of $10^{-4}$ for most kernels considered in the VLIDORT
BRDF supplement [*Spurr*, 2004]. However, at and near the hotspot region, many more quadrature
points and Fourier terms ($N_{FOURIER}$) will be needed, as we will demonstrate below. Indeed, Lorente
et al. (2018) found that in order to reach an accuracy of $10^{-3}$ over the hotspot region, 720 Gaussian
points were needed for the azimuth integration and 300 Fourier terms for the reconstruction of any
BRDF in terms of its Fourier components; they also determined that, in the final implementation
of the surface BRDF in the DAK radiative transfer model (Doubling–Adding KNMI, [Lorente et
al., 2017]) designed to perform with optimal simulation time, some 100 Fourier terms and 360
Gaussian points were necessary for proper hotspot characterization.
These values of $N_{BRDF}$ and $N_{FOURIER}$ are still unacceptably high, and in order to use VLIDORT to
simulate the hotspot signature with a modest number of discrete ordinates, we have made an
empirical modification to the hotspot correction in the RossThickHT-M model by using the
function $sin^x(\zeta) * \frac{1}{sin^x(\zeta_0)}$ to replace $\zeta/\zeta_0$, where $x = 2 + \sin(\theta'_r)$. Thus:
$$K_{vol} = \frac{4}{3\pi} \frac{\left(\frac{\pi}{2}-\zeta\right)\cos\zeta + \sin\zeta}{\cos\theta'_r + \cos\theta'_i} (1 + \frac{1}{1 + sin^x(\zeta) * \frac{1}{sin^x(\zeta_0)}}) - \frac{1}{3}.$$ 
(11)



We use the nomenclature RossThickHT-X to indicate the model with the hotspot correction given
in Eq. (11).
In the next section, we first examine the above sets of hotspot signatures, with particular emphasis
on the accuracy of reconstructed BRDFs in terms of the two numerical indices $N_{BRDF}$ and $N_{FOURIER}$.
We then determine the impact of a scattering atmosphere, using these hotpot BRDF quantities as
inputs to VLIDORT calculations based on standard-atmosphere pressure/temperature profiles with
Rayleigh scattering and aerosols in the form of an optically-constant layer from the surface to 6.5
km having total optical depth of 0.2; aerosol optical properties are taken from a "continental
pollution" aerosol type [Hess et al., 1998], with lognormal poly-disperse size distribution.





### *3.* **Results and Discussion**


*3.1 Hotspot Comparisons and BRDF reconstruction accuracy*


Figure 1 shows a comparison of the volume-scattering kernel for the three hotspot models,
RossThickHT-X, RossThickHT-C and RossThickHT-M, with actual hotspots at three different
solar zenith angles in the principal-plane backscatter direction. For reference, the original
RossThick kernel is also plotted. Hotspot peaks from the three models are the same, and the hotspot
peak is higher and narrower at larger zenith angles. Major differences between the three models
are outside the hotspot region. As indicated by [Jiao et al., 2013], one asset of RossThickHT-C is
that it better matches the RossThick model in regions beyond the hotspot, while on the other hand,
there remain some differences between the RossThickHT-M and RossThick model away from the
hotspot. Our new model RossThickHT-X has the same advantage as RossThickHT-C, in that
agreement with the standard RossThick model beyond the hotspot region is accurate.

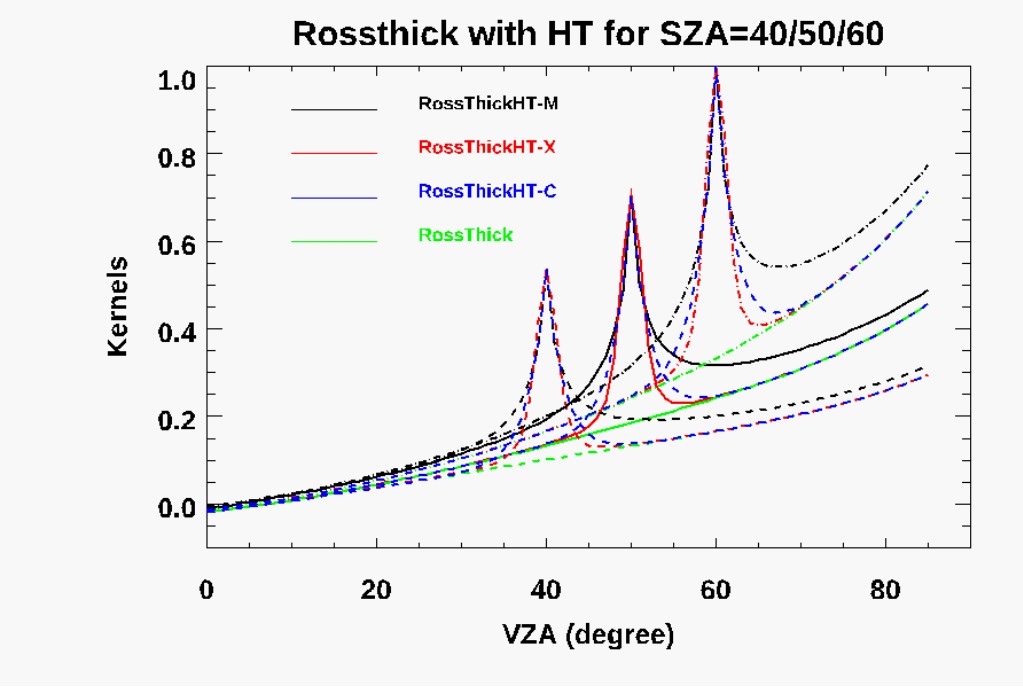


**Figure 1**. Four Ross-Thick volume scattering kernels for a range of reflection zenith angles,
and for three solar incident angles as indicated; reflectance is in the principal plane.


The major advantage of our new hotspot correction model is the rapid convergence for
reconstruction. Table 1 lists values of $N_{BRDF}$ (number of azimuth quadrature abscissae) and
$N_{FOURIER}$ (number of Fourier Terms) that are needed to reconstruct the BRDF to different accuracy
levels; the accuracy is computed as the relative difference of the reconstructed BRDF to its exact
value at the hotspot. Compared to numbers required for the RossThickHT-M, values of $N_{BRDF}$ and
$N_{FOURIER}$ for the RossThickHT-X case are 10 to 60 times smaller (Table 1). These results show
that RossThickHT-X converges much faster than RossThickHT-M. We see also that convergence





of RossthickHT-C is somewhat faster than that for RossThickHT-M but still much slower than
that for RossThickHT-X.
**Table 1**. Values of $N_{BRDF}$ and $N_{FOURIER}$ needed to reconstruct a hotspot with $\zeta_0 = 1.5°$.

| | | **RossThickHT-M** | | **RossThickHT-X** | | **RossThickHT-C** | |
|---|---|---|---|---|---|---|---|
| # | Accuracy (%) | NBRDF | N_FOURIER | NBRDF | N_FOURIER | NBRDF | N_FOURIER |
| 1 | 1 | 2810 | 1402 | 278 | 139 | 1578 | 789 |
| 2 | 0.5 | 5620 | 2807 | 324 | 162 | 3158 | 1579 |
| 3 | 0.4 | 7020 | 3509 | 338 | 169 | 3948 | 1974 |
| 4 | 0.3 | 9360 | 4679 | 356 | 178 | 5264 | 2632 |
| 4 | 0.2 | 14040 | 7019 | 382 | 191 | 7896 | 3948 |
| 5 | 0.1 | 28080 | 14039 | 428 | 214 | 15794 | 7897 |

While both numbers are necessary for the reconstructed BRDF accuracy, the main impact comes
from the number of Fourier terms $N_{FOURIER}$ used, when the value of $N_{BRDF}$ is twice (or more) that
of $N_{FOURIER}$. In Figure 2, using a fixed value $N_{BRDF} =100$ for the RossThickHT-M, RossThickHT-
C and RossThickHT-X models, we show the dependence of the relative error of the reconstructed
BRDF on the solar zenith angle for four different values of $N_{FOURIER}$. Choices of $N_{FOURIER}$ (23, 31,
63 and 95) correspond to values 12, 16, 32 and 48 for the number $N_{STREAMS}$ (number of half-space
polar discrete ordinates) used in VLIDORT ($N_{FOURIER} = 2N_{STREAMS} - 1$). In this example, also used
by [Lorente et al., 2018] (their Figure 6), the BRDF represents a vegetated surface over Amazonia
at wavelength 758 nm with free parameters $[P_1, P_2, P_3] = [0.36, 0.24, 0.03]$ taken from MODIS
band 2 (841–876 nm) to account for the increase in surface reflectivity near 700 nm.
Overall, the error decreases with increasing values of $N_{FOURIER}$. The error also increases with those
viewing angles at which the hotspot occurs, since the hotspot peaks are higher and narrower for
larger viewing angles. Errors for all three models are large when $N_{FOURIER}$ is as small as 23. The
advantage of RossThickHT-X starts to show when $N_{FOURIER}$ increases to 31, but this is not
significant when the hotspot viewing angle is larger than 45°. When $N_{FOURIER}$ is set to 95, the
performance of RossThickHT-X is much better than that for the other two models; the error is less
than 1% even for large viewing hotspot angles, whereas the corresponding errors using
RossThickHT-M or RossThickHT-C are still at the 5-8% level for hotspots at viewing angles
larger than 30°. Overall, the error with RossThickHT-C is slightly smaller than that for
RossThickHT-M.

**Figure 2**. Accuracy of Fourier-reconstructed BRDFs relative to their exact values, for the three Ross-Li models. $N_{BRDF} = 100$, with $N_{FOURIER}$ set to four different values as indicated. Surface BRDF parameters represent a vegetated surface over Amazonia at 758 nm, with $[P_1, P_2, P_3] = [0.36, 0.24, 0.03]$.

Next we examine simulated TOA reflectances at 758 nm with the three hotspot models providing inputs to the main VLIDORT RT calculations. We again set $N_{BRDF} = 100$ and $N_{STREAMS} = 12, 16, 32$ and 48. Results are shown in Figure 3 for two solar zenith angles. The hotspot signature is evident at 30° (upper panels) and 50° (lower panels), and the peak signature with aerosols present is higher than that without aerosol. The widths of the hotspots in Figure 3 are very similar, confirming the argument of [Powers and Gerstl, 1988] that the hotspot width is expected to be relatively invariant to atmospheric perturbations. Lines of different colors correspond to simulations using different values of $N_{STREAMS}$; in general, differences between these lines are pretty small, especially in the



atmosphere without aerosol and when the viewing angle is less than 60°. To better illustrate
patterns in TOA reflectance values using different values $N_{STREAMS}$, we used the simulated
reflectances obtained with $N_{STREAMS}$ = 48 as the reference, and the results of this comparison are
shown in Figure 4.

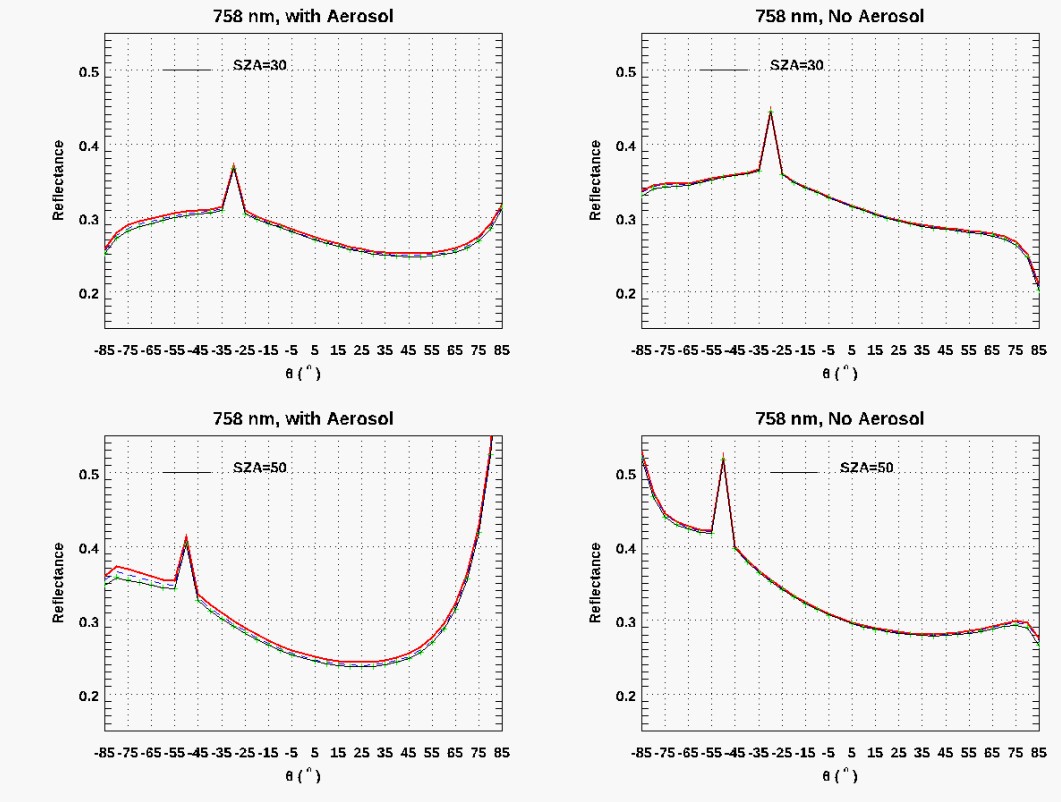


**Figure 3**. TOA reflectance as a function of viewing zenith angle, simulated by VLIDORT at 758
nm with a Ross-Li surface BRDF model with hotspot correction RossThickHT-X. Geometries
are in the principal plane for two solar zenith angles as indicated, and results were obtained with
and without aerosol. Surface BRDF parameters represent a vegetated surface over Amazonia at
758 nm with $(P_1, P_2, P_3)$ = (0.36, 0.24, 0.03). The red solid line represents the simulation $N_{STREAMS}$
= 12, blue dashed line is for $N_{STREAMS}$ = 16, with the remaining lines for $N_{STREAMS}$ = 32 (green)
and $N_{STREAMS}$ = 48 (dark); the latter two lines are almost aligned.

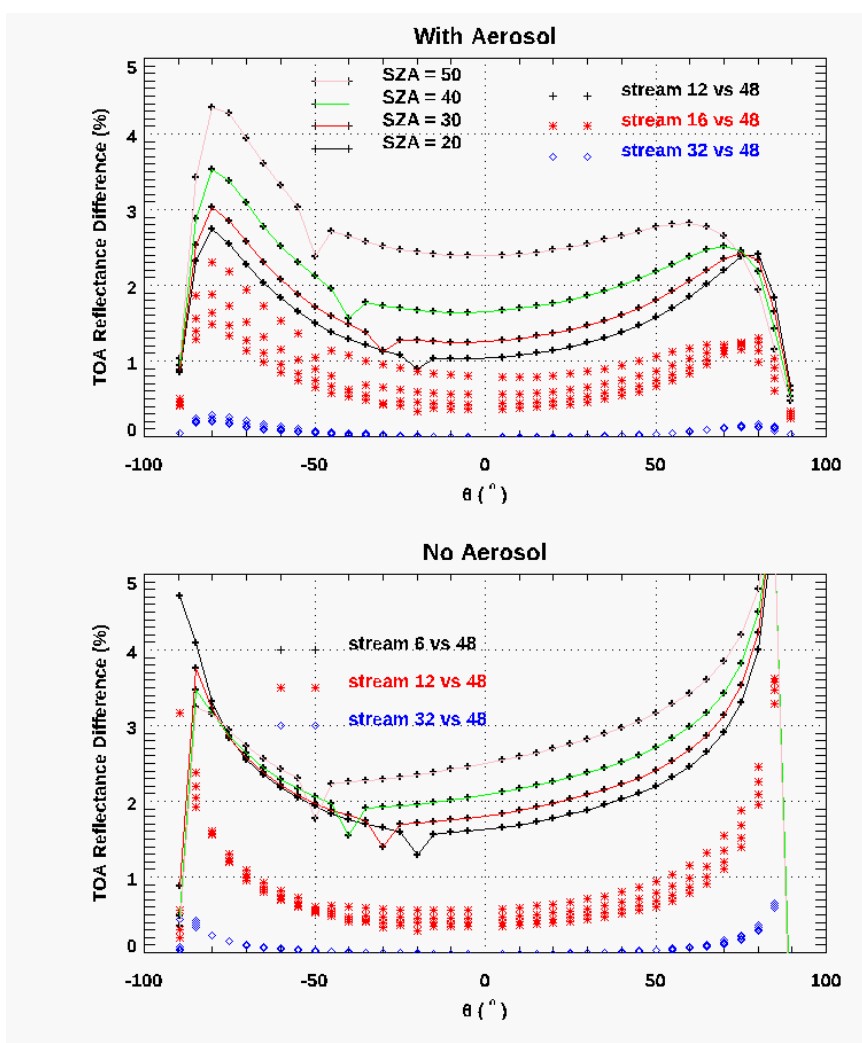

365
366

**Figure 4**. Same set-ups as Figure 3, but now plotting the TOA reflectance differences with four
solar zenith angles as indicated.

From Figure 4 it is evident that relative differences in TOA reflectances for an atmosphere with
aerosols are larger than those for the atmosphere without aerosols. As the typical viewing angle
range for BRDF kernels is mostly within 60°, we will focus on these differences for viewing angles
$< 60°$. In the upper panel we see that TOA differences (comparing $N_{STREAMS} = 12$ with $N_{STREAMS}$
$= 48$) increase with solar zenith angle; the difference at SZA = 50° is almost double than that at
SZA = 20°. The relative difference in percentage at the hotspot region is smaller than beyond
hotspot, which is easy to understand as the absolute value of the TOA reflectance at the hotspot is
larger. In both cases with and without aerosol, TOA reflectance differences (comparing $N_{STREAMS}$





= 32 with $N_{STREAMS}$ =48) are very small; VLIDORT simulations with $N_{STREAMS}$ = 32  are accurate
enough in this case.
For the atmosphere with aerosol, the bias in simulated TOA reflectances using $N_{STREAMS}$ = 16
(relative to $N_{STREAMS}$ = 48) is 0.5-1.0%.  In the clear atmosphere without aerosol, the bias of using
$N_{STREAMS}$ = 6 can be in the region 2-3%, but the bias with $N_{STREAMS}$ = 12 is around 0.5%,
suggesting that the setting for $N_{STREAMS}$ should be 12 or higher in a Rayleigh atmosphere overlying
a hotspot surface.
As noted already, $N_{FOURIER}$ = 2$N_{STREAMS}$−1.  Compared to the value of $N_{FOURIER}$ needed for
reconstruction of surface BRDFs near the hotspot (Table 1), that is, $N_{FOURIER}$ = 139-162 for an
accuracy of 0.5-1.0%, the values of $N_{FOURIER}$ = 23 (for the Rayleigh scenario) and $N_{FOURIER}$ = 63
(for the atmosphere with aerosol) needed for full VLIDORT RT simulations are much smaller.
The reason for this reduction lies with the separation in VLIDORT between the first order (FO:
single scattering and direct reflectance) calculations and the multiple-scatter (MS) calculations in
VLIDORT. The first-order calculation in VLIDORT is always done with full accuracy with solar
beam and line-of-sight attenuations treated for a curved atmosphere, and with an exact value for
the surface BRDF used to calculate the "direct-bounce" reflectance (which is very often the
dominant contribution from the surface). No Fourier reconstruction is necessary for this
contribution. For the MS contribution, multiple scatter is treated using Fourier cosine/sine azimuth
expansions and associated Fourier terms for both the truncated phase matrix for scattering and the
diffuse-field BRDF contributions. The important point to note here is the use of the exact BRDF
for the direct-bounce contribution in VLIDORT; RT models without this FO/MS separation will
be constrained by the need to use a Fourier-expanded reconstruction for the direct-bounce BRDF
contribution.
The results shown in Figures 3-4 are confined to a single standard atmosphere and aerosol model.
In the next section below, we use VLIDORT simulations to investigate the impact of scattering on
hotspot signatures. For this study, we choose $N_{BRDF}$ = 200 and $N_{STREAMS}$ =32; this should be
conservative enough to avoid any uncertainty associated with the use of surface BRDFs and the
choice of stream numbers in VLIDORT.
### 3.2. Impact of scattering on the hotspot signature at TOA
Here we use the three parameters ($P_1$, $P_2$, $P_3$ ) =  (0.0399. 0.0245, 0.0072) for the RTLSR surface
BRDF model. These are the spatially averaged parameters from MODIS (BRDF/albedo product
MCD43A1) band 3 (459–479 nm) over Amazonia (latitude 5° N – 10° S, longitude 60 – 70° W)
for March 2008 [Lorente et al., 2018]. TOA reflectances are calculated as a function of viewing
zenith angle in the principal plane, with the solar zenith angle set at 30° (Figure 5).  In this
experiment, two calculations are plotted, one using the new hotspot correction model,
RossThickHT-X, and the other using the RTLSR BRDF model without a hotspot correction. From
Figure 5 it is clear  that the TOA-hotspot signature at 469 nm is very small, likely due to the
influence of stronger Rayleigh scattering. The addition of aerosol scattering further reduces the
hotspot signature at SZA 30° and it is hard to discriminate the TOA reflectance difference between
the runs with and without hotspot correction. This observation agrees with the results from [Bréon
et al., 2002], in which it was noted that no significant hotspot signature has been observed when
the surface reflectance is very small, as in the blue channel or over the ocean. For the longer
wavelength at 645 nm, the TOA-hotspot signature is obvious, and the addition of aerosol scattering
reduces the hotspot signature slightly compared to the situation with molecular scattering only.



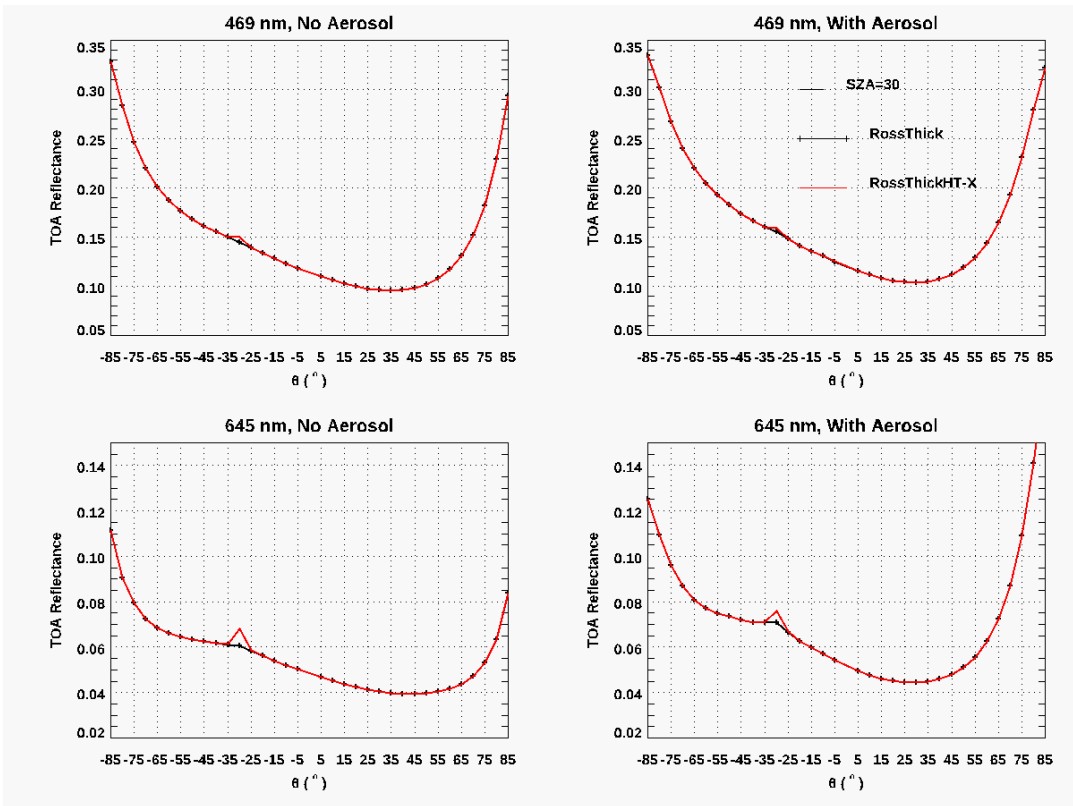



**Figure 5**. VLIDORT TOA reflectances as a function of viewing zenith angle with solar angle 30° in the principal plane, at 469 and 645 nm using a Ross-Li surface BRDF model with hotspot correction RossThickHT-X , and with and without aerosol. The aerosol model used is the same as in Figure 3, with optical depth 0.2. Surface BRDF parameters represent a vegetated surface over Amazonia with $(P_1, P_2, P_3)$ = (0.0399. 0.0245, 0.0072).

We also examine the hotspot signatures in 765 and 865 nm, two wavelengths used in POLDER data analysis. The three linear weighting parameters in the BRDF model are $(P_1, P_2, P_3)$ = (0.36, 0.24, 0.03), which is the same set as that used by [Lorente et al., 2018]. As noted already, these are taken from MODIS band 2 (841–876 nm) to account for the "red-edge" increase in surface reflectivity near 700 nm (e.g. [Tilstra et al., 2017]). To test the representativeness of band 2 at 758 nm, Lorente et al.[2018] scaled the parameters from band 3 (459–479 nm) using the ratio of reflectances at 772 nm and 469 nm; they found that differences with parameters taken from MODIS band 2 were negligible. Since we would like to focus on the difference of the impact of atmospheric scattering on the hotspot signatures at 758 and 865 nm, we have chosen to use the same two sets of surface BRDF parameters. The results are plotted in Figures 6 and 7. To highlight the differences caused by the $3\pi/4$ factor normalizing the volume-scattering kernels $K_{vol}$ (see note in Section 2.2), we have added in Figure 7 two simulated TOA reflectances, one based on the original hotspot correction model from Maignan et al. [2004] (RossThickHT-M) and the other using the BRDF noted in the paper of Lorente et al. [2018] (indicated by "RossThickHT-L").


Compared to Figure 5, much larger TOA-hotspot signatures at both 865 and 758 nm are evident
in Figures 6 and 7 respectively, and they are slightly larger at SZA=50° than at SZA=30°. As
expected, in the scattering region beyond the hotspot (±5º), the TOA reflectance using
RossThickHT-X agrees very well with that using the original RossThick model. However, from
Figure 7, we see that the simulated reflectance using RossThickHT-M is slightly larger than that
using RossThick model even in a region of ±15° beyond the hotspot, particularly in the large
viewing angles in the forward direction. In the region of ±5° to ±15° beyond the hotspot, the
simulated reflectance using RossThickHT-M is clearly larger than that using RossThick and
RossThickHT-X.

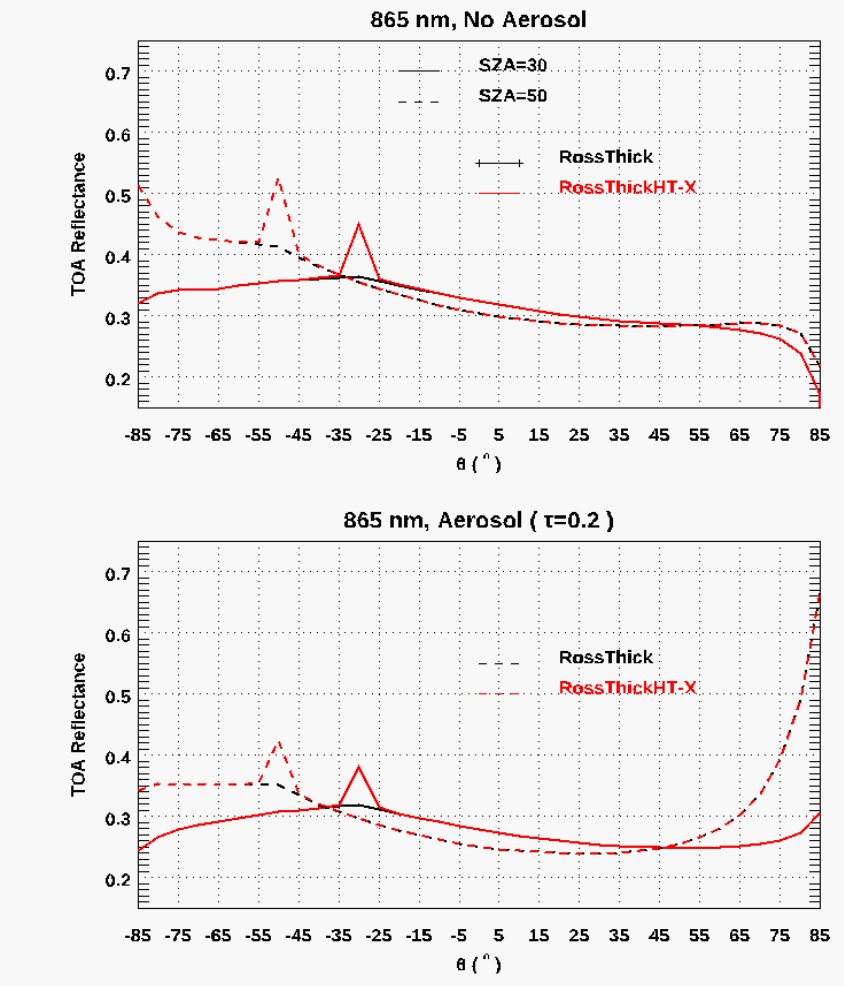


**Figure 6**. Same as Figure 5 but results are calculated at 865 nm for solar zenith angles 30° and
50°. Surface BRDF parameters represent a vegetated surface over Amazonia with $(P_1, P_2, P_3) =$
455 (0.36, 0.24, 0.03).



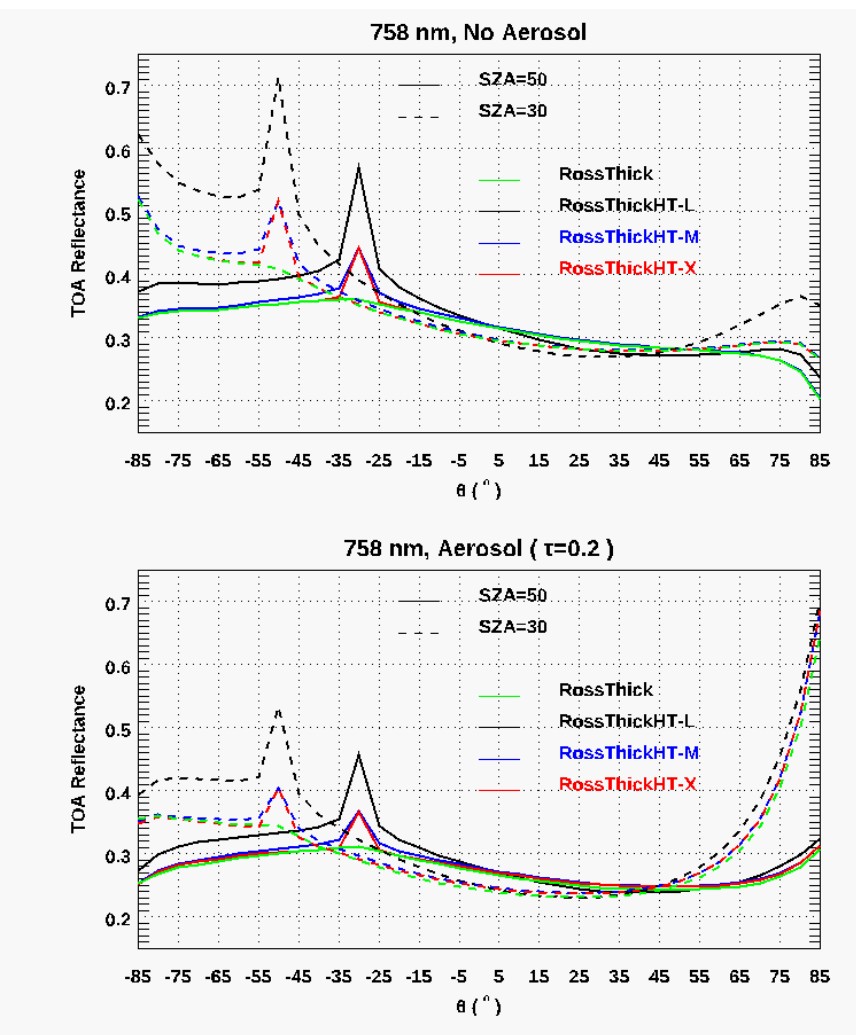


**Figure 7**. Similar to Figure 6 but results calculated at wavelength758 nm. For comparison, we
have added simulated TOA reflectances using the original hotspot correction model from
[Maignan et al., 2004] (RossThickHT-M) and again using the model in [Lorente et al., 2018],
which is a factor of $4\pi/3$ times larger than RossThickHT-M in the hotspot region and is denoted
here as RossThickHT-L.

To better quantify the hotspot effect and the impact due to scattering in the atmosphere, we define
the "hotspot amplitude" as the difference between the TOA reflectance at the hotspot and the
corresponding TOA reflectance calculated without hotspot correction, namely:

$$HT_{Amplitude} = \frac{R\,(\theta o, \theta, \varphi = 180, RossThickHT - Li)}{R(\theta o, \theta, \varphi = 180, RossThick - Li)}.$$

The impacts of molecular and aerosol scattering on these amplitudes are illustrated in Figure 8 for
a range of hotspot viewing angles and for four wavelengths. For comparison, the hotspot



amplitudes at the surface are also plotted. From Figure 8, it is evident that scattering in the
atmosphere smooths out the hotspot signature at TOA, and the impact of scattering is much larger
in the visible compare with that in the near-infrared part of the spectrum. Even in the visible, the
amplitude of the hotspot signature at 469 nm is much smaller than that at 645 nm. Similarly, the
amplitude in 758 nm is smaller than that at 865nm. These simulated results agree well with the
analysis of POLDER data by [Bréon et al., 2002]; at 440 nm, they found that the amplitude of the
hotspot signature is very small, confirming that the atmospheric contribution to the reflectance
increase at the backscattering direction is negligible. The much larger amplitudes observed at 758
nm and 865 nm also confirm the findings by [Maignan et al., 2004], who showed that near-infrared
measurements are preferred to those in the visible, not only because of the larger-amplitude
directional effects but also because of the lower atmospheric perturbation.  Indeed, Maignan et
al.[2004] suggested that near-infrared measurement data is better suited for the evaluation of
different BRDF models. From Figure 8 we can also see that in the near-infrared, the amplitude of
the hotspot signature increases with the zenith angle (right panel); however, the angular
dependencies in the surface hotspot and the TOA hotspot are almost opposite in the visible,
especially for an atmosphere without aerosols.

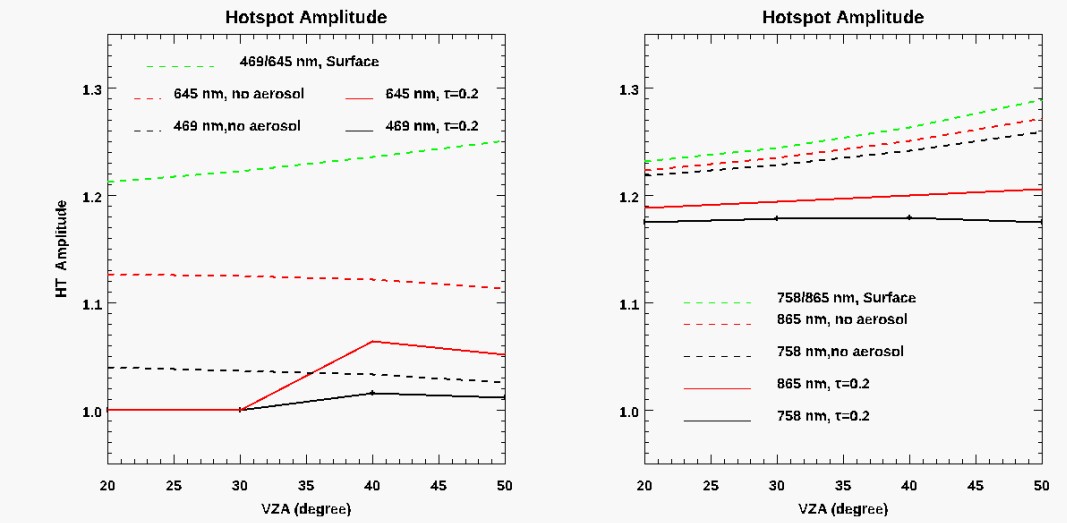


**Figure 8**. Comparison of hotspot amplitudes at the TOA for an atmosphere with and without
aerosols in visible (469 and 645 nm, left panel) and near-infrared (658 and 865 nm, right panel).
Hotspot amplitudes at the surface are computed using the differences between the RossThickHT-
Li and RossThick-Li BRDF models.
In the processing of POLDER data done by [Bréon et al., 2002] and [Maignan et al., 2004], only
molecular scattering to first order was taken into account for the atmospheric correction. As there
is no correction for the effects of aerosol scattering or the coupling of surface reflectance with
molecular scattering, absolute values of the reflectances may not be fully representative of the
surface for POLDER [Bréon et al., 2002] .  From our simulations shown in Figure 8, the amplitude
of the hotspot signature with aerosol scattering included is smaller than that without aerosol,
suggesting that the results from POLDER [Bréon et al., 2002] might underestimate the amplitude
of hotspot signature at the surface.



A final issue is related to a factor difference that exists between the equation of [Lorente et al.,
2018] (i.e. their Eq. A1) with our Eq. (8), which is the one used in [Maignan et al., 2004]. The one
used by [Lorente et al., 2018] is $3\pi/4$ times larger; this discrepancy results in a TOA-hotspot
signature more than twice as large, as shown in Figure 7. This factor difference is the main reason
that the TOA-hotspot signatures shown by [Lorente et al., 2018]their Figure 5) at 469 and 645 nm
are higher than the ones observed in this paper. In addition, as seen in [Lorente et al., 2018] (their
Figure 5), the lower TOA-hotspot signature generated by LIDORT (as opposed to those from the
other two RTMs) is likely due to the deployment of an older version of LIDORT that did not
include the hotspot correction. From the upper panel of Figure 7, it is evident that the hotspot peak
using RossThickHT-L seems too high, particularly for the hotspot occurring at 50°. From the
analysis of POLDER data, Bréon et al.[2002] found that the hotspot reflectance amplitude is
generally of the order 0.10 – 0.20 at 865 nm and 0.03 – 0.18 at 670 nm, although the full range of
values is wide. Therefore, we think that the use of an equation with a factor of $3\pi/4$ discrepancy is
likely to overestimate the hotspot effect, and we caution users to be careful to check the equations
for the presence of this $3\pi/4$ factor, even when using MODIS BRDF products.



## 4. Summary and Conclusions

In remote sensing, it is common practice to deploy a simple kernel-driven semi-empirical model with three free parameters to represent land surface BRDFs (excepting snow and ice); the best model is the RossThick/LiSparse combination with a correction to account for the hotspot [Maignan et al., 2004]. In our study, we modified this BRDF model to improve convergence of the Fourier azimuth series decomposition, and using this new hotspot correction, we further studied the impact of scattering on the atmospheric hotspot signature using the VLIDORT RTM.

With the improved hotspot correction, we found that the numbers of Gaussian points ($N_{BRDF}$) and Fourier Terms ($N_{FOURIER}$) are more than 10 times smaller than those needed with the original hotspot model from Maignan et al. [2004]; this makes our BRDF model much more practical for use with VLIDORT to simulate the hotspot signature at the TOA. We also showed the new hotspot model agrees very well with the original RossThick model away the hotspot region, thus allowing the use of this single model in the condition with and without hotspot.

We carried out a number of investigations on the impact of molecular and aerosol scattering on the hotspot signature at the TOA. TOA reflectances were calculated for different solar and viewing angles and at four wavelengths; the main findings from our study are:

1.  In agreement with previous analysis using POLDER measurement data, hotspot signatures in the near-infrared are larger than those in the visible.
2.  Also in agreement with the POLDER study, the hotspot amplitudes at TOA and the surface both increase with solar zenith angle in the near-infrared; however, at 469 and 645 nm, this increase with solar zenith angle is not obvious at TOA.
3.  Scattering by molecules and aerosols in the atmosphere tends to smooth out the hotspot signature at TOA, and the hotspot amplitude is reduced when aerosols are added to an otherwise clear (Rayleigh scattering only) atmosphere. These results also showed that atmospheric scattering does not generate hotspot-like signatures and does not change the width of the BRDF-induced hotspot.
4.  In VLIDORT, the direct-beam solar reflectance is calculated using the exact BRDF (rather than in a truncated Fourier-series form); this means that smaller values of $N_{FOURIER}$ (i.e. 23 and 63 for atmospheres without and with aerosol scattering) can be used in for the multiple scattering calculations in VLIDORT to obtain hotspot signature with acceptable accuracy.

Atmospheric corrections in the POLDER data processing were performed using Rayleigh-only single scattering without any consideration of aerosol. Our simulations suggest that the amplitude of hotspot signature at the surface is likely underestimated in the analysis of hotspot signature using POLDER data [Bréon et al., 2002].

Another issue related to the hotspot correction in the model used by Lorente et al. [2018] is the scaling by a factor of $3\pi/4$; this may lead to the amplitude of hotspot too high in large solar zenith angle. It is recommended that users take care to check the kernel equations when using the three parameters from MODIS BRDF products to generate the BRDF.

Our results highlight the importance of the including aerosol scattering in the retrievals of surface BRDF (hotspot). The agreement between our simulated results and observations from POLDER measurements enhances our understanding of the nature of the hotspot and the impact on it by atmospheric scattering. It is also clear that VLIDORT makes accurate simulations of the hotspot



effect, and the results obtained here can be used as benchmarks. Our improved hotspot kernel is
now a standard feature in the latest version of the VLIDORT BRDF supplement code.

**Description of author's responsibilities**

XX, XL and RS conceived of the idea. XX and RS led the writing. All authors edited the
manuscript.

**Funding**

This research was supported by NASA SBG program.

**CRediT authorship contribution statement**

**Xiaozhen Xiong:** Methodology, Writing – original draft, Formal analysis, Investigation. **Xu**
**Liu:** Funding acquisition, Supervision, Writing – review & editing, Conceptualization. **Robert**
**Spurr:** Methodology, Writing – review & editing, Formal analysis. **Ming Zhao**: Coding,
Analysis. **Wan Wu, Qiguang Yang, Liqiao Lei:** Writing – review & editing.

**Declaration of Competing Interest**

The authors declare that they have no known competing financial interests or personal
relationships that could have appeared to influence the work reported in this paper.

**Acknowledgements**

This research was supported by the NASA SBG program and JPL. Resources supporting
this work were provided by the NASA High-End Computing (HEC) Program through the NASA
Advanced Supercomputing (NAS) Division at NASA Ames Research Center.



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
