# Peer review of "An Improved BRDF Hotspot Model and its Use in VLIDORT to Study the Impact of Atmospheric Scattering on Hotspot Directional Signatures in the Atmosphere"

_Atmospheric Measurement Techniques, 2023_

## Author Comment (AC1)

We appreciate your thorough review and helpful comments. We have accepted all your editorial comments and made corrections in the revision of the manuscript. The following are our one-to-one responses to your comments/questions. ***Please note that our responses will be given in Bold Italic type face.***

General Comments from reviewer: The so-called Hot Spot is the increased in surface reflectance as the observation geometry nears backscatter. It has a significant impact over a few degree around this particular direction. Note that the Hot Spot is difficult to observe on the ground as the Hot Spot is then hidden by the observer own shadow. In this paper, the authors modify an existing analytical model of the BRDF, including the Hot Spot, so that it can be efficiently used by the VLIDORT radiative transfer modeling that uses Fourier expansion. As such, it should be made clear that the "improvement" of the model is not in its ability to reproduce the truth but in its ability to be used by LIDORT.

*Answer: We agree that our contribution is a numerical improvement in BRDF Hotspot Modeling for radiative transfer models that use Fourier expansions to characterize the azimuthal dependence of the BRDF. Specifically, our hotspot model reduces the number of quadratures needed to integrate the Fourier terms, which results in improvements in both the speed and the numerical accuracy. We will validate this improved model with real data in a future paper. We have added these points clearly in the abstract and conclusion.*

Having the ability to model the Hot Spot in VLIDORT is certainly a useful feature that could be used in the community and I therefore welcome the efforts of the author in that direction. However, as describe below, the paper does not demonstrate that the modeling is accurate and reproduces the specific signatures of the Hot Spot. In particular,

- The modified model does reproduce a "Hot Spot like" signature, but this signature is significantly different than the original modeling by Maignan and Breon that has been validated against observations. Thus, it may not be appropriate for those who need an exact representation of the Hot Spot angular signature.

*Answer: Thanks for your good suggestion. You are correct that our modified model can reproduce the hot spot signature close to the backscatter direction and the hotspot magnitude (at exact backscatter), but there are some differences between our model and that of Maignan and Breon when scattering angles are near to the hotspot. In the abstract and conclusion, we have added a sentence that it may not be appropriate for those who need an exact representation of the Hot Spot angular signature.*

- the figures in the paper not have the angular resolution that is necessary to properly investigate the impact of atmospheric scattering on the Hot Spot width, which is one of the claimed results of the paper.

*Answer: Considering the difference between our model with that of Maignan and Breon at scattering angles close to the hotspot, we will remove all statements and discussion about the effect of atmospheric scattering on hot spot width in the paper.*

Therefore, significant improvements of the papers are needed before it can be published

*Answer: Significant improvements will be made to revise the paper following the above suggestions; this will include replots of all figures.*

Detailed comments

In the paper, the authors seem to agree that the RossThickHT-M modeling is the reference as it has been validated against POLDER data in its ability to reproduce the amplitude but also the angular shape of the Hot Spot. The authors have developed a modified version, RossThickHT-X for its ability to be efficiently developed with a Fourier expansion. However, Figure 1 shows that the two models have an angular shape that is significantly different. Such differences may not be acceptable for those who have a specific interest in the Hot Spot signature. The "X" version of the model may certainly be acceptable for some application that just need to show some kind of a Hot Spot, but not for all applications. Thus the "improved" qualification to this new model is only numerical. This should be made clear. Also, the differences shown in Figure 1 should be discussed.

*Answer: We have added a statement in the abstract and the conclusion to highlight the limitation of this model. We have also replotted Figure 1 and added a discussion of the differences between the RossThickHT-X with RossThickHT-M models., as follows:*

*"For model RossThickHT-X, the angular shape around the hot spot peak (VZA=SZA) is not so sharp as the reference model RossThickHT-M, thus, it may not be appropriate for those who need an exact representation of the hot spot angular signature. However, from limited validation Jiao et al. [2013] found that RossThickHT-M apparently overestimates the hot spot magnitude. A visual examination to Figure 2 of Jiao et al. [2013] shows RossThickHT-M looks too sharp near hotspot peak as well."*

One conclusion from the study is that the Hot Spot Width is not affected by the atmospheric scattering. This is deduced from the results shown in Figure 3. However, this angular resolution of this figure is by far insufficient to analyze the width of the Hot Spot. The statement must be removed or this figure improved.

*Answer: Yes, we removed this statement about the effect of atmospheric scattering on the hot spot width. We also re-plotted Figure 3 after increasing the number of BRDF calculations within a 2$^o$ spread around the actual hotspot.*

Also, there is insufficient information about the hypothesis on the aerosol load that are used in the simulations

*Answer: At the end of Section 2, we have revised the manuscript as follows:*

*"We then determine the impact of a scattering atmosphere on TOA reflectances at hot spot angles. The VLIDORT RT model was used to calculate TOA reflectances, based on these hot spot BRDF signatures and with US standard-atmosphere pressure/temperature profiles as input. Two cases are studied - one with Rayleigh scattering only and the other with aerosols. Aerosol is assumed to be uniformly distributed from the surface to 3.0 km with the total optical depth of 0.2 at 550 nm, and aerosol optical properties are taken from a "continental pollution" aerosol type [Hess et al., 1998], with a lognormal poly-disperse size distribution. Use of an optical depth of 0.2 for aerosol represents a moderately polluted atmosphere but might be a little higher than that for background aerosol."*

The paper makes several statements about the amplitude of the Hot Spot as a function of wavelength. However, the results they obtain (ie an amplitude that is smaller in the visible than in the near IR) is entirely a consequence of their hypothesis on the surface reflectance. Obviously, if the amplitude of the HS (at the surface) is large, it is also so at the TOA. I strongly suggest (i) to add the surface reflectance (at the surface) on Figure 5 and (ii) to cleary indicate in the text how much this result is because of the lower amplitude is because of the lower reflectance and because of the increased atmospheric scattering

*Answer: Thanks for these very helpful suggestions. We have added surface reflectance in Figure 5. We revised the 1$^{st}$ paragraph in Section 3.2 accordingly. Please see the following for detailed revisions:*

*" From the comparison of TOA reflectances at all angles between the left and the right panels in Figure 5, we can see that the TOA reflectance in the atmosphere with aerosol is overall larger than that without aerosol, an indication that the aerosol scattering increases the TOA reflectance. Compared to the case with molecular scattering only, the addition of aerosol leads to an increase of TOA reflectance near hot spot peak by ~ 8% and 17% at 469 and 645 nm respectively.*

*However, from a comparison of the TOA reflectances for surface with hotspot and without hotspot, i.e. using RossThickHT-X and RossThick, we found that at 469 nm the increase of surface reflectance at hot spot results in an increase of TOA reflectance by ~ 4% and 2% for the no-aerosol and with-aerosol conditions, respectively, and at 645 nm the amounts of increase are ~12.5% and 7% accordingly. These results indicate that for the longer wavelength at 645 nm, the TOA-hotspot signature is much stronger than at 469 nm, due to the smaller Rayleigh scattering at 645 nm, and the inclusion of aerosol scattering further smooths out the hotspot signature at the TOA by about 40-50% compared to the atmosphere with molecular scattering only."*

Atmospheric scattering reduces the Hot Spot amplitude. This is shown in Figure 8. At the first order, the amplitude is reduced by a factor exp(-to m) where m is the air mass computed from the

solar and view zenith angles, and to is the optical depth. I strongly suggest to (i) compare the result of Figure 8 with this simple estimate, and (ii) discuss in the text .

*Answer: Here is what we estimated:*

*When SZA increases from 20º to 50º and for condition with aerosol, the HS amplitudes at 469 nm and 645 nm decrease by -1.08% and -2.14%, and at 758 nm and 865 nm the HP amplitudes increase by 0.03% and 1.52% respectively. By using the first order approximation and for atmosphere with aerosol, the HS amplitude can be estimated as [1.-exp(-tau/cos(SZA)) + alb_ht\*exp(-tau\*M)]/ [1-exp(-tau/cos(SZA)) + alb0\*exp(-tau\*M)], where M =1/cos(SZA)+1/cos(VZA)), Alb0 and Alb_ht are the reflectances of no-hotspot and with-hot spot. With this first-order calculation, when SZA increases from 20º to 50º the HS amplitude would decrease by -0.24% and -0.29% at 469 and 645 nm, and at 758 nm and 865 nm the HS amplitudes increase by 0.96% and 1.63% respectively. So, overall the first-order approximation agrees with VLIDORT simulations reasonably well, and their difference can be attributed to the impact by multiple scattering. However, if it is estimated using a factor of exp(-τ\*M), as reviewer suggested, the HS amplitudes decrease by -34.3% and -21.3% at 469 and 645 nm, and by -15.0% and -11.8% at 758 and 865 nm respectively. The problem is such calculation is that it has not considered the downwelling radiance and the difference of surface reflectance between no-hotspot and with-hotspot and its change with SZA.*

*Below is what we added in the context:*

*"When the SZA increases from 20º to 50º, the HS amplitude at 469 nm decreases by -1.34% and -1.08% for atmospheric conditions without aerosol and with aerosol, respectively. The HS amplitude at 645 nm decreases by -1.24% and -2.14% similarly. In contrast, the HS amplitudes increase by 3.36 (0.03)% at 758 nm, and by 3.9 (1.5) % at 865 nm as SZA increases from 20º to 50º. Since molecular scattering is much smaller than in the visible, the large difference in the amounts of HS amplitude increase between no-aerosol and with-aerosol conditions indicates the impact of multiple scattering, and the existence of aerosol smooths out the TOA hotspot signature. The increase of HS amplitudes with SZA following with the increase of surface reflectance in the near infrared, particularly in the no-aerosol condition, indicates that the HS amplitude is largely affected by surface reflectance in the near infrared."*

Also, I see that the computations are made for an optical depth of 0.2. It may be worthwhile to mention that this is rather high and that most satellite observation are less affected by the aerosols than this simulation may suggest.

**Answer: We agree, and we have added the recommended statement.**

On line 495 it is said that the HS amplitude in Bréon et al might be underestimated. By how much is that ? A factor of 2 or 1% ? Obviously, the impact would not be the same.

*Answer: We give an estimate based on the difference between the conditions with and without aerosol. Here is what we have added:*

*"Based on the differences of the HS amplitudes between the atmosphere with aerosol and without aerosol, we estimate that, on average, the HS amplitude is underestimated by 4.0 ±1.7% when not considering aerosol for a moderately polluted atmosphere with optical depth of 0.2."*

Also, the authors comment on the fact that different studies use different version of the BRDF kernels that differ by a factor 3p/4. They claim that this explain difference between their results and those from others. However, a Kernel is always used associated with a weight. Obviously, when one uses a version of a Kernel that differs by a constant factor, the weight must be adjusted accordingly. In fact Kernel models are mostly used to fit remote sensing measurements. Using a Kernels that differ by a constant multiplying factor just leads to a different inverted weight, withno impact on the retrieved BRDFs. The current text is somewhat misleading on this subject and must be corrected

*Answer: We agree that there is no need to emphasize this, so we have revised this paragraph and removed those sentences that may lead to confusion. We also removed this subject from the abstract and conclusion.*

Regarding the conclusions:

We also showed the new hotspot model agrees very well with the original RossThick model away the hotspot region, thus allowing the use of this single model in the condition with and without hotspot.

It should be clear that the new hotspot model does not agree well with the corrected "Hot Spot" model close to the Hot Spot direction.

*Answer: We have made this clear in the manuscript. We also revised the first finding as follows:*

*In agreement with previous analyses using POLDER measurement data, hotspot signatures in the near-infrared are larger than those in the visible.*

This is a conclusion that only derives from the hypothesis made on the surface reflectance.

*Answer: we added "…as it is less impacted by molecular scattering, since the hotspot amplitude is reduced when aerosols are added to an otherwise clear atmosphere".*

It should be made clear whether this reduction is by the direct transmission factor, or by a different value.

**Answer:** *This reduction is not due solely to the direct transmission factor, as we have shown by comparing the first order estimation and the corresponding calculation using VLIDORT will full accounting for multiple scattering in the atmosphere.*

These results also showed that atmospheric scattering does not generate hotspot-like signatures

Some authors have claimed that there may be coherent backscatter.  As this potential feature is not accounted for in Vlidort, the modeling is not really a demonstration of its inexistence in the true world.  In addition, some aerosol models show an increase of the phase function close to the 180 scattering angle.  Such aerosols would generate an Hot Spot like signature, but may not have been tested in the author simulatiThe figures shown in the paper cannot be used to make that conclusion

*Answer: We have removed this sentence from the conclusion: "It is also clear that VLIDORT makes accurate simulations of the hotspot"*

 There is no validation in that paper to make that statement.  In fact, if one trust the RossThickHT-M modeling, as it has been somewhat compared to POLDER observations, then Figure 1 indicates that the modified model (RossThickHT-X) makes an inaccurate simulation of the HotSpot

*Answer: We agree that there is no validation, so we have removed this sentence in last paragraph:*

As discussed above, the differences between RossThickHT-M and RossThickHT-X indicate that the Hot Spot signature is uncertain for those who have a specific interest in the Hot Spot signature.  Therefore, the RossThickHT-X cannot be used as benchmarks

*Answer: we removed the sentence regarding benchmarking.*

Our improved hotspot kernel is now a standard feature in the latest version of the VLIDORT BRDF supplement code. It should be made clear that the improvement is only to increase the numerical efficiency, and not to obtain more accurate simulation of the Hot Spot signature

 *Answer: The last paragraph was revised as follows*

*"Our improved hotspot kernel is now a standard feature in the latest version of the VLIDORT BRDF supplement code that has significantly improve the numerical efficiency. Since this new model has not been validated using any real observation data and considering the difference between this model and the original hotspot model from Maignan et al. [2004] in scattering angles close to hot spot, it may not be appropriate for those who need an exact representation of the hot spot angular signature around the hot spot point."*

Other comments

Line 19 : *almost* coincide

*Answer: Changed*

Line 20-23: This sentence is wrong.  The RTLSR model is analytical and does not require, therefore Fourier expansion.  Only its use within models such as Vlidort does.

*Answer: added "for its use coupled with radiative transfer modelling (RTM)"*

Line 24 : Not clear what "converge" refers to here

*Answer: Convergence here refers to the summation of the Fourier-azimuth expansion of the radiation field – if the addition of a further Fourier term changes the overall field by less than a certain very small convergence criterion, then the Fourier-azimuth RT expansion has "converged".*

Line 38 : attenuation => attention

*Answer: we corrected this.*

Line 65 : What is the "forward scatter direction" ?

*Answer: In the forward scatter direction, the scattering angle is zero.*

At many places, reference Vermont et al [2009] is used.  This ref does not exist and is probably Vermote et al [2009]

*Answer: We rechecked the references of Vermote et al.(2009)  and corrected them.*

Line 204 : The equation seems to be for the phase angle, not the *scattering* angle

*Answer*: *Yes, $\zeta$ is the phase angle.*

Line 264 : it => It

*Answer: changed*

Line 270 : numerical accuracy of $10^{-4}$ . Is that an absolute accuracy (in which case, what is the unit) or a relative accuracy ?

*Answer: it is the relative accuracy.*

---

## Author Comment (AC2)

We appreciate your thorough review and helpful comments. We have accepted all your editorial comments and made corrections in the revision of the manuscript. The following are our one-to-one responses to your comments. Please note that our responses will be given in Bold Italic type face.

General comment from reviewer: This study presents an alternative model of the hotspot correction that was designed to improve numerical computation efficiency. The new formulation then is compared with the conventional hotspot correction approach for the Reciprocal-Ross-Li BRDF model in the VLIDORT package. With the new development, authors also examined the impact of atmospheric scattering on the simulated hotspot behavior as observed from the TOA.

I appreciate that authors provided a detailed and interesting history of the BRDF developments and RTM modeling developments. The manuscript was well written and results were presented with substantial discussions. Although this is an interesting study, the work needs a substantial improvement in its focus and conclusions.

The innovative part of the paper is the presentation of a new hotspot correction formulation (Eq 9) of computational advantage. This formulation is similar to that proposed by Bréon et al. [2002] but replacing scattering angle by  $sin^x(scatter angle)$ . However, authors didn't provide why the sine function is introduced here.

**Answer: We have added the following:**

"... by choosing the function with a smooth transition near the hotspot peak and considering that sin ( $\zeta$ ) can be used to replace  $\zeta$  approximately when the phase angle is small value. We experimented with different powers of this function, finally coming up with the factor x as noted in the text. ...

Comparing with other hotspot correction functions, the proposed new function requires less number of streams and Fourier terms to achieve a similar accuracy. How about the computational efficiency? Would this contribute a significant improvement in practical satellite remote sensing algorithms?

Answer: Lorente et al. (2018) found that, in order to reach an accuracy of 10-3 over the hotspot region, 720 Gaussian points were needed for the azimuth integration and 300 Fourier terms for the reconstruction of any BRDF in terms of its Fourier components. These figures are impractical for applications using radiative transfer models such as DISORT and Doubling–Adding; in the end, Lorente et al. (2018) used 360 Gaussian points and 100 Fourier terms for their calculations. In VLIDORT (as is also the case with the DISORT model), the computation time goes roughly as the third power of the number of streams. Since the number of terms used in our hotspot model is more than 10 times less than that specified for the original hotspot model (as shown in the Table 1), there would be a considerable performance gain with the BRDF simulations using RT models such as DISORT, Doubling–Adding and VLIDORT.

Above Table 1 we added:

"the computation time goes roughly as the third power of the number of streams. Since the number of terms used in our hotspot model is more than 10 times less than that specified for the original hotspot model (as shown in the Table 1), there would be a considerable performance gain with the BRDF simulations

With the new development, authors examined the impact of atmospheric scattering on the simulated hotspot behavior as observed from the TOA. From several simulation experiments, authors come up 4 major findings that are listed in the Summary and Conclusion section. Three of those findings, however, were demonstrations of previous studies or known physics. That means, they can be demonstrated even without the development of the new hotspot correction function. So, these analysis should not be the focus of this study.

**Answer: We revised the phrase "The main findings from our study are" with "These simulations using VLIDORT show that"**

505-511: A same set of BRDF weight factors were used for simulations in Figure 7 for various hotspot correction approaches. The results showed the RossThickHT-L (Lorente et al., 2018) approach were higher than other methods due to the lack of the 4/3pi term. Authors concluded that the hotspot peak using RossThickHT-L seemed too high. I don't think it is a reasonable conclusion. Because it is not appropriate to apply a same set BRDF weight factors to different approaches at the first place. Without constraints (fitting with) observations, such comparisons make no sense.

Also, MODIS BRDF products can be only applied to the Ross-Li-Reciprocal model to construct the BRDF, because the retrieval (i.e., observation fittings) were based on this model. Direct application of them to a different model (no matter which hotspot correction for the Ross-Li-Reciprocal model) would have a chance to deviate from the observation constraints.

Answer: We agree with this comment. In the paper of Lorente et al., 2018, the authors obtained the VLIDORT result using an older version of the code, and this result showed the hotspot peak that was smaller than that generated with the other RT models. We think the reason for this lies with a scaling factor difference between the hotspot BRDF equation cited in Lorente et al., 2018 and the equation used in the earlier VLIDORT model. Hence, we have added this simulation results here in order to bring attention to users when using scaling factor data from the MODIS BRDF product.

We agree that there is no need to emphasize this, so we revised this paragraph and removed the sentences that may lead to confusion. We also removed corresponding wording from the abstract and conclusion.

The major finding #2 is not reasonably explained by the authors. The trends of hotspot amplitudes with respect to solar zenith angle at 469 and 645 nm are different from that in the near-infrared; this difference is mainly due to the stronger Rayleigh scattering in the shorter wavelengths, which is more pronounced for longer path length at larger solar zenith angles.

Answer: We agree and have added the following "... due to stronger Rayleigh scattering at shorter wavelengths, which is more pronounced for longer path lengths at larger solar zenith angles."

Specific comments

Line 81: An expansion is needed for RTLSR at its first appearance. *Answer: This has been added*

Figure 8: Should the x-axis be Solar zenith angle? Answer: The solar zenith angle equals the viewing zenith angle at the exact hotspot point.

Line 514: It might be arbitrary to say "the best model is the ....". How about "the commonly used model is the ..."

Answer: We have adopted this wording.

---

## Author Response (AR2)

The following comments have been addressed.

(1) The optical depth 0.2 was applied to all the studied spectral wavelengths in the paper. Authors may want to clarify this doesn't consider the spectral dependence of optical depth of atmospheric aerosols.

*Response: added in L304-305"*
*", and it doesn't consider the spectral dependence of optical depth of atmospheric aerosols"*

(2) Figure 2: The use of Nfour in the plot title to represent N_Fourier is misleading. Please use a different term.

*Response: changed Nfour to N_FOURIER to make it consistent with Table 1.*

(3) Figure 3 and Figure 5: please swap the columnar panels in either Figure 3 or Figure 5 to make them consistent, i.e, 'with aerosol' and 'no aerosols' are current on different sides.

*Response: Figure 3 was replotted by swap two panels, and changed the title in the right panel. Figure 5 was replotted by changing the title "with aerosol to 'Aerosol (\tau=0.2)'*
*in the right panel.*

(4) Same as above, but for Figure 4 and Figure 6-7.

*Response: Figure 4 was replotted by changing the title "with aerosol to 'Aerosol (\tau=0.2)' in the lower panel, and it is consistent with Figures 6-7 now.*

(5) Figure 3 to figure 8, please use a consistent annotation for aerosols with optical depth of 0.2. For instance, Figure 6 and 7 currently use 'Aerosol (\tau=0.2)', Figure 8 uses 'aerosol(0.2)', whereas other figures simply use 'With Aerosol'.

*Response: All were changed as requested. In Figure 8, we still used no aerosol and with aerosol, and in the Figure caption we add "For aerosol, the optical depth is set to 0.2."*